# Evaluation and Selection of De-Icing Salt Based on Multi-Factor

**DOI:** 10.3390/ma12060912

**Published:** 2019-03-19

**Authors:** Guoju Ke, Jun Zhang, Bo Tian

**Affiliations:** 1Department of Civil Engineering, Tsinghua University, Beijing 100084, China; kgj17@mails.tsinghua.edu.cn; 2Research Institute of Highway Ministry of Transport, Beijing 100088, China; tbb73@yahoo.com

**Keywords:** de-icing ability, concrete peeling amount, plant drying rate, analytic hierarchy process, optimal item

## Abstract

De-icing salts can greatly ease traffic congestion but introduce corrosion of concrete and damage to plant growth. The decision of which de-icing salt to use becomes a crucial issue. In this study, several representative de-icing salts were investigated, and the effects of de-icing ability, salt freezing corrosion on concrete, and plant growth were comprehensively tested. Finally, the decision of de-icing salt was made based on analytic hierarchy process (AHP). Results show that de-icing salts achieving the best de-icing effect are not the same at different concentrations. De-icing salts of 3% concentration have the greatest corrosion to concrete. Notably, magnesium chloride and calcium magnesium acetate have the least impact on plants among all studied de-icing salts. Using AHP, ethylene glycol and calcium magnesium acetate are selected as optimal items under different priorities.

## 1. Introduction

Frozen and slippery roads cause traffic congestion and accidents and endanger the safety of people and vehicles. Statistics show that the probability of traffic accidents in ice and snow is 4–5 times that in normal weather. The use of de-icing salts can reduce accidents caused by snow and ice by 88.3% [1].

De-icing salts can ease traffic congestion but introduce problems that cannot be ignored, such as corrosion of roads and bridge concrete, impact on plant growth, and damage to groundwater and soil. Under the action of freezing and thawing, the damage of concrete and the severity of some salt corrosion are approximately ten times those of ordinary corrosion. The service life of roads affected by de-icing salt may be decreased by more than 50% [2,3]. Under the action of de-icing salt, the salt concentration of groundwater becomes high, which causes physiological drought of plants; sodium ions cause abnormal absorption of plant nutrient elements; the accumulation of toxins in plants destroys their normal metabolism, which harms the growth of street trees, isolation belts, and lawns; these phenomena make de-icing salts a “green killer” [4,5,6,7,8]. In the spring of 2005, the use of de-icing salt in the urban area of Beijing resulted in the death of 11,000 pavement trees, 1.5 million shrubs, 200,000 m^2^ of lawns, and economic losses of more than 30 million Yuan [9]. De-icing salts can also cause groundwater pollution [10,11,12], thereby affecting the life of water and plants [13] and even harming human health.

In the 1970s, China began to use de-icing salts. On 6 February, 2006, the amount of de-icing salt used in Beijing reached 0.73 million tons due to special weather. At present, finding a suitable de-icing salt is crucial and urgent because of the high emphasis on resource conservation and environmental protection.

The analytic hierarchy process (AHP) was first proposed by Professor Saaty in the University of Pittsburgh in the 1970s [14]. This decision-making method is simple and practical; a model of decomposition, judgment, and synthesis based on hierarchical analysis solves multi-objective and multi-criteria problems [15]. In this study, we use the AHP to comprehensively consider the ability of de-icing salt to melt snow, the corrosion of concrete, and the influence on plants under three factors and obtain the optimal items among the alternative de-icing salts.

## 2. Materials and Methods

### 2.1. Materials

De-icing salts have many types. According to composition, de-icing salt can be divided into inorganic, organic, and mixed types. De-icing salt can also be classified into chlorine salt type (e.g., sodium chloride, calcium chloride, and magnesium chloride), non-chlorinated salt type (e.g., organic or inorganic alcohol and ammonia), and mixed type (e.g., mixed chlorine and non-chlorinated salts). This study selects eight kinds of conventional de-icing salts in literature, namely, sodium chloride, magnesium chloride, calcium chloride, calcium magnesium acetate, urea, sodium formate, potassium acetate, and ethylene glycol [16,17,18], and one type that is commonly used in Beijing (Ji-suo II) (Table 1). The first eight are chemically pure reagents, and Ji-suo II was purchased from a Beijing company.

### 2.2. Methods

#### 2.2.1. Melting Test of De-Icing Salts

In accordance with the provisions of the “Snow Melting Agent” (GB/T 23851-2017) [18], 25 mL of de-icing salt solution was prepared with concentrations of 18% and 29% and poured into a 50 mL beaker. Then, the beaker was placed in a low-temperature test chamber ((−10 ± 1) °C) to freeze for 12 h until use.

In ten plastic boxes of the same diameter and height, 100 mL of water was added. Then, they were placed in a low-temperature incubator at (−10 ± 1) °C for icing. After icing, each plastic box with ice cubes was removed. The water and ice on the outer wall of the box were wiped prior to weighing. Thereafter, the spare de-icing salt was poured into the box with ice cubes, and the box was placed into the low-temperature incubator again. After 30 min, the box was removed, the liquid was poured out, and the quality of the remaining ice and the box was weighed and recorded. After weighing, the liquid was poured back into the box with ice cubes, which was placed in a cryostat. The boxes were removed at 60, 90, 120, and 150 min, separately, and the remaining ice and boxes were weighed after pouring the liquid therein.

#### 2.2.2. Salt Freezing Corrosion Test of Concrete under the Action of De-Icing Salt

Conventional pavement cement concrete contains cement of 360 kg·m^−3^, sand of 626 kg·m^−3^, stone of 1270 kg·m^−3^, water of 144 kg·m^−3^, and water reducing agent of 3.6 kg·m^−3^. The test mold was made of PVC (Polyvinyl chloride) pipe with an inner diameter of 200 mm and a height of 80 mm. The fresh concrete was loaded into the test mold and vibrated, and wood was used to wipe the molded surface slurry. The resultant was placed into the curing room for standard curing, as shown in Figure 1. The test box used in the test was made of stainless steel, and a non-metallic mat as placed on the bottom of the container. The mat did not absorb water, did not deform when immersed in water, and did not affect the solution during the test. After the test specimen reaches the required age, the molded surface was immersed in a certain concentration of de-icing salt solution, and the immersion depth was 4–6 mm, as shown in Figure 1.

The salt-freezing equipment used DYD-1 concrete single-sided salt freezing equipment, as shown in Figure 2. The intelligent computer control system can control the limit temperature in the tank (testable range from 20 °C to −20 °C). The freeze-thaw system was operated as follows: each freeze–thaw cycle of the concrete test block was completed within 12 h, which starts from cooling down to −20 °C for 4 h and then holding for 3 h and heating to 20 °C for 4 h and then holding for 1 h. Each cycle was repeated ten times, and peeling amount of concrete is collected and weighted while replacing the new salt solution.

#### 2.2.3. Effect of De-Icing Salt on Plant Growth

In flowerpots of same sizes (43 cm × 19 cm × 14 cm), the same grass species were planted. The selected grass species were mainly easy to operate, have a certain representativeness, and have a short test period. When the growth was prosperous and roughly the same, 25 mL of certain concentrations of No. 1 to No. 9 de-icing salts and No. 10 water was sprayed. After spraying, the growth was observed. The yellowing of the plant tip after spraying various de-icing salts was recorded, as shown in Figure 3. Dry day refers to the number of days when the tip of the plant begins to turn yellow; the dry rate refers to the proportion of plants that turn yellow.

## 3. Results and Analysis

### 3.1. De-Icing Ability

#### 3.1.1. De-Icing Ability Test under 18% Concentration of De-Icing Salt

Table 2 shows that the cumulative amount of de-icing of the nine de-icing salts increases with the increase in de-icing time, De-icing accumulation refers to the amount of ice that can be melted by de-icing salt over a certain period of time. The maximum value is reached at 120 min, and the amount of de-icing decreases with the increase in de-icing time after 120 min. The reason is that the concentration of de-icing salt affects the level of freezing point. A high concentration of de-icing salt corresponds to a low freezing point and strong melting effect of de-icing salt. By contrast, a small concentration of de-icing salt indicates a high freezing point. Therefore, the ideal de-icing effect is not achieved when the temperature is low. “Snow Melting Agent” (GB/T 23851-2017) [18] requires that the ability to melt snow and ice is not less than 90% of the capacity of sodium chloride. Thus, potassium acetate does not meet the requirements and is unsuitable as a de-icing salt.

#### 3.1.2. De-Icing Ability Test under 29% Concentration of De-Icing Salt

In accordance with the standard “Snow Melting Agent” (GB/T 23851-2017), eight different de-icing salts other than potassium acetate are selected to determine the ability under 29% concentration for comparing the effects of de-icing salt at different concentrations.

As shown in Table 3, all the de-icing salts reach the maximum amount of de-icing at 60 min, but the amount decreases at 90 min. This result is due to the fact that the concentration of de-icing salt affects the level of freezing point. A high concentration of de-icing salt implies a low freezing point and strong melting effect of de-icing salt. On the contrary, a small concentration of de-icing salt corresponds to a high freezing point. Therefore, the ideal de-icing effect is not achieved when the temperature is low. At 29% concentration, the sodium formate solution has crystallized after 12 h of freezing in the low-temperature test chamber at −10 °C.

### 3.2. Salt Freezing Corrosion of Concrete under the Action of De-Icing Salt

From Table 4, the following results are obtained:(1)Under 3% concentration of de-icing salt, the peeling amount of concrete surface increases with the increase in the number of cycles. Under the action of five de-icing salts, namely, sodium chloride, ethylene glycol, urea, sodium formate, and calcium chloride, the peeling amount of concrete is high. Among the five de-icing salts, ethylene glycol, sodium formate, and urea have higher salt freezing corrosion on concrete than sodium chloride. By contrast, magnesium chloride and calcium magnesium acetate have less salt corrosion damage to concrete.(2)Under 10% concentration of de-icing salt, the peeling amount of concrete surface increases with the increase in the number of freeze-thaw cycles. Among the eight de-icing salts, the peeling amount of concrete with sodium chloride, ethylene glycol, urea, and sodium formate is more serious. Ethylene glycol and sodium formate are more corrosive to concrete than sodium chloride. On the contrary, calcium magnesium acetate, calcium chloride, and magnesium chloride have less salt corrosion damage to concrete.(3)Under 20% concentration of de-icing salt, the peeling amount of concrete surface is extremely small. In particular, urea, calcium magnesium acetate, and magnesium chloride have not peeled off the surface of concrete after 30 freeze-thaw cycles.

The peeling amount of concrete is compared after 30 freeze-thaw cycles under 3%, 10%, and 20% concentrations of de-icing salt. When the concentration of de-icing salt is 3%, the surface of concrete is highly exfoliated. When the concentration of de-icing salt is 20%, the surface of concrete barely peels off after 30 freeze-thaw cycles.

### 3.3. Effect of De-Icing Salt on Plant Growth

As shown in Figure 4 and Figure 5, the plants sprayed with water are prosperous and show no dryness on the seventh day after spraying of the solution. The dry rate of plants sprayed with calcium magnesium acetate is 6%, which is the lowest group of plants in the experimental group. This result indicates that calcium magnesium acetate has the least damage to plants among the nine de-icing salts. The dry rate of plants sprayed with sodium chloride is 22%, that of plants with magnesium chloride, calcium chloride, and Ji-suo II is less than 22%, and that of plants with urea, ethylene glycol, sodium formate, and potassium acetate is more than 22%. In other words, these de-icing salts are more harmful to plants than sodium chloride.

As shown in Figure 6 and Figure 7, the plants sprayed with 10% sodium chloride and potassium acetate have the earliest occurrence time of yellowing phenomenon. Therefore, the two de-icing salts have the fastest damage to plants. The tip of the plant sprayed with calcium magnesium acetate is the last to show the yellowing phenomenon (i.e., on the fifth day). This finding shows that calcium magnesium acetate, although has certain damage to plants, has the least damage to plants compared with other de-icing salts.

After 7 days of spraying with de-icing salt, all plants are dry. The dry rate represents the damage of the final de-icing salt to the plant. As shown in Figure 7, the plants sprayed with 10% concentration of calcium magnesium acetate, magnesium chloride, and calcium chloride have lower dry rate than those sprayed with sodium chloride. Therefore, the three de-icing salts are more harmful to plants than sodium chloride. Ethylene glycol, urea, sodium formate, and potassium acetate are also slightly more harmful to plants than sodium chloride.

Figure 8 shows that when the concentration of de-icing salt is 10%, the dry rate of plant is greater than that under 3% concentration. This result indicates that high concentration of de-icing salt is more harmful to plants than low concentration of de-icing salt. Moreover, 3% and 10% concentrations of de-icing salt show a consistent trend of damage to plants. The effects of ethylene glycol, urea, sodium formate, and Ji-suo II on the dry rate of plants at seventh day are greater than the effect of sodium chloride. The effects of calcium magnesium acetate, magnesium chloride, and calcium chloride on the rate are less than the effect of sodium chloride. The dry rate of plants at the seventh day is the lowest under the actions of calcium acetate and magnesium. Therefore, the effect of calcium magnesium acetate on plant growth is the weakest.

## 4. Comprehensive Evaluation of De-Icing Salt by Analytic Hierarchy Process

### 4.1. Establishment of a Hierarchical Analysis Structure

According to the Chinese standard “Snow Melting Agent” (GB/T 23851-2017), the ability of de-icing is the most important indicator in the reference standard. Damage to plants and salt damage to concrete are the indicators that can be specified depending on needs. In determining the alternative de-icing salt, the de-icing salt with a de-icing ability lower than 90% of that of sodium chloride is removed, and the unknown component of Ji-suo II is removed. The remaining seven de-icing salts are selected and sorted by combining test indicators.

Following previous analysis, the de-icing ability of 18% concentration of de-icing salt in 30 min is used as the de-icing ability index. The peeling amount of concrete with 3% concentration of de-icing salt under 30 freeze–thaw cycles is the corrosion index of concrete. The dry rate at seventh day of 3% concentration of de-icing salt sprayed on plants is used as an indicator of the influence on plants. The hierarchical analysis structure is constructed and shown in Figure 9. 

### 4.2. Establishment of Judgment Matrixes

After the hierarchical analysis model is established, the elements of each layer are compared by two to construct a comparison judgment matrix. The judgment matrix is constructed by the commonly used a 1–9 scale method. The size of the judgment value represents the importance of the corresponding column element compared with the row element. A large judgment value indicates high importance of the column element, as shown in Table 5.

The values of 2, 4, 6, and 8 indicate that the influence of the i-th factor relative to the j-th factor is between the above-mentioned two adjacent levels, a_ij_ = 1/a_ji_.

Criteria layer: When using different de-icing salts, the weight of de-icing ability, the salt freezing corrosion of concrete, and the influence on plant growth should be considered. The weight of de-icing salt should also be considered to satisfy the three criteria. The priority considered between the three criteria is different in dissimilar situations. When the ability of de-icing is the most important consideration, the influence on plant is second, and the corrosion of concrete is the least important, the judgment matrix A-B (a) is as shown in Table 6a. When the ability of de-icing is as important as the impact on plant, followed by the corrosion of concrete, a different judgment matrix A-B (b) is obtained as shown in Table 6b.

Scheme layer: In this optimization, sodium chloride is used as the reference de-icing salt. The de-icing amount, peeling amount, and drying rate are 100. The corresponding indexes of other de-icing salts are converted on the basis of the test results, as shown in Table 6. The values of the three indicators of the seven de-icing salts are equally divided into nine equal parts, and the importance degree is calibrated. As shown in Table 7, a large amount of de-icing indicates an excellent performance; the maximum interval value is 9, and the minimum interval value is 1. Moreover, a small amount of peeling implies an excellent performance; the minimum interval value is marked as 9, and the maximum interval value is marked as 1. Furthermore, a small rate of dryness rate corresponds to an excellent performance; the minimum interval value is marked as 9, and the maximum interval value is marked as 1. After the pair wise comparison, the judgment matrix is constructed (e.g., the C1 calibration value is 6, the C2 calibration value is 9, C_12_ is 1/4, and C_21_ is 4). From the analysis, the scheme layer judgment matrices B1-C, B2-C, and B3-C are obtained, as shown in Table 8, Table 9 and Table 10.

### 4.3. Consistency Check and Hierarchical Ordering

Constructing a judgment matrix mathematically solves complex problems and simplifies the analysis of the problem. The method of consistency check usually first calculates the maximum eigen value λ_max_ and the weight eigenvector W of the judgment matrix and then determines whether the judgment thinking is consistent using CI = (λ_max_ − n)/(n − 1) [13,14]. When the value is 0, the judgment matrix has complete consistency. The consistency of the judgment matrix can be determined by the average random consistency index RI of the judgment matrix and the judgment matrix of first to ninth order. The values of RI are as shown in Table 11.

The random consistency ratio is used to measure whether the judgment matrix has satisfactory consistency. This ratio is the ratio of the consistency index of the judgment matrix to the same-order average random consistency index. When CR = (CI)/(RI) < 0.10, the judgment matrix has satisfactory consistency; otherwise, the judgment matrix is adjusted until it has satisfactory consistency.

### 4.4. Hierarchical Total Ordering

In accordance with the judgment matrix of the criterion and scheme layers, the results are combined to obtain a hierarchical integrated ranking, as shown in Table 12. Considering the different weights of the three indicators, different A-B judgment matrices and weight values and hierarchical comprehensive sorting conclusions are obtained.

### 4.5. Decision Making

After the total weight value of each de-icing program for the total target is obtained, it is divided into three levels according to the total weight value of de-icing salt. The evaluation type with a weight value greater than or equal to 0.2 is excellent, that is, grade A. The evaluation type with a weight value greater than or equal to the evaluation type of 0.1 and less than 0.2 is good, that is, grade B. The evaluation type of a weight value less than 0.1 is qualified, that is, grade C.

When the ability of de-icing is the most important consideration, the impact on plants is second, and the corrosion of concrete is the third, the total weight values of each de-icing salt are 0.124, 0.222, 0.036, 0.183, 0.033, 0.194, and 0.208. The de-icing salts of grade A are ethylene glycol and calcium chloride; those of grade B are sodium chloride, calcium magnesium acetate, and magnesium chloride; the rest is grade C with ethylene glycol as the preferable item.

When the ability of de-icing and the impact on plants are important concerns, the total weight values of each de-icing salt are 0.119, 0.182, 0.039, 0.221, 0.032, 0.211, and 0.195. The de-icing salts of grade A are calcium magnesium acetate and magnesium chloride; those of grade B are sodium chloride, ethylene glycol, and calcium chloride; the rest is grade C with calcium magnesium acetate as the preferable item.

## 5. Conclusions


(1)At 18% concentration of de-icing salt, the de-icing accumulation of all de-icing salts increases as de-icing time progresses. The maximum value is obtained at 120 min, and then the amount of de-icing decreases. At 29% concentration, the amount of de-icing reaches the maximum at 60 min and decreases at 90 min.(2)When the concentration of de-icing salt is 3%, the surface of concrete is highly exfoliated. When the concentration of de-icing salt is 20%, the surface of concrete is barely peeled off after 30 freeze-thaw cycles.(3)At 10% concentration, the dry rate of plants is greater than 3%. Moreover, 3% and 10% concentrations of de-icing salt show a consistent trend of damage to plants.(4)Seven de-icing salts are tested under de-icing at 18% concentration, salt peeling at 3% concentration, and plant dry rate at 3% concentration. A hierarchical analysis model is established on the basis of the priority of the three criteria, and a comparative judgment matrix is constructed. A de-icing salt that meets the requirements is selected. The preferred de-icing salts differ for dissimilar priorities of the three criteria.


## Figures and Tables

**Figure 1 materials-12-00912-f001:**
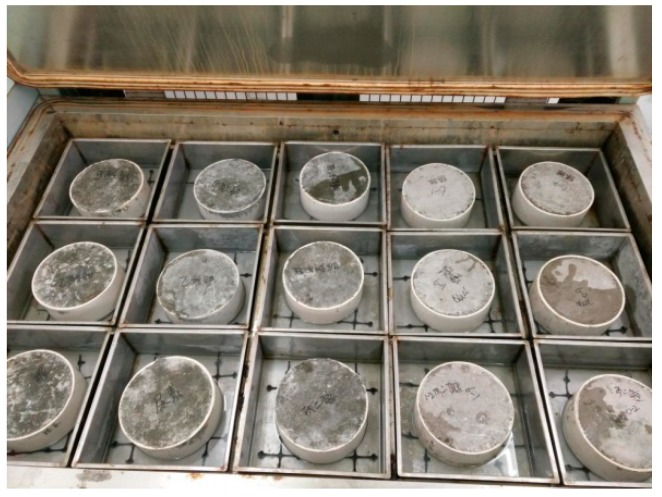
Preparation of specimen.

**Figure 2 materials-12-00912-f002:**
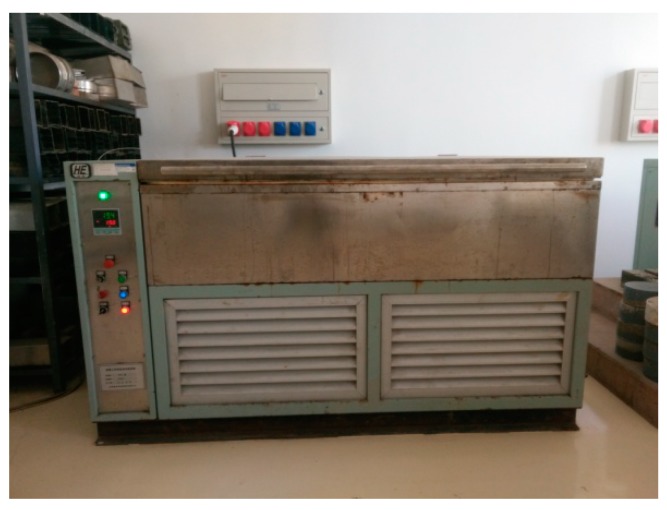
DYD-1 Concrete salt freezing machine.

**Figure 3 materials-12-00912-f003:**
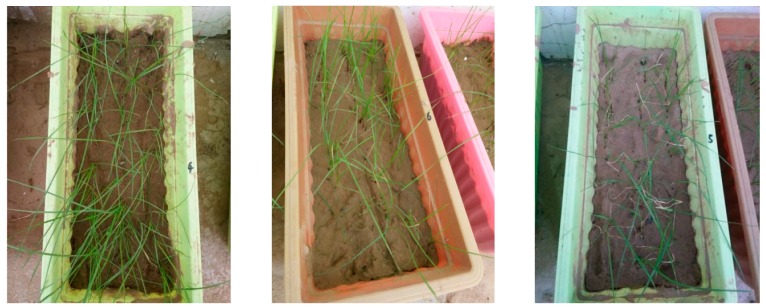
Process of plant tip yellowing.

**Figure 4 materials-12-00912-f004:**
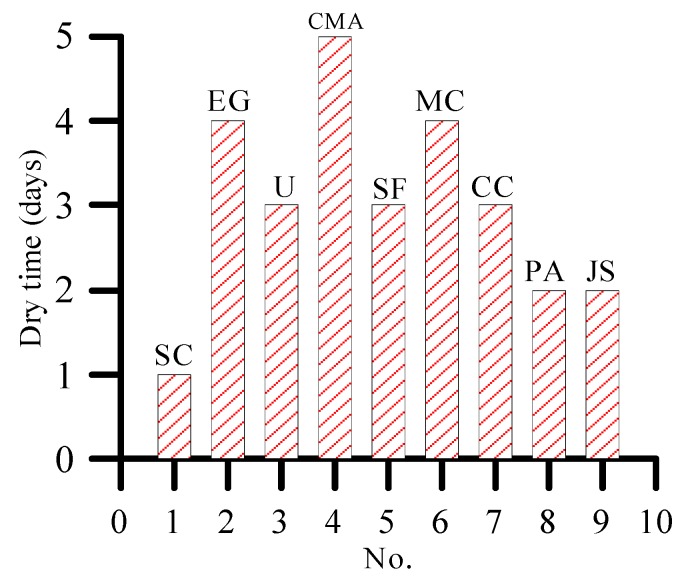
Plant tip yellowing days under 3% concentration of de-icing salt.

**Figure 5 materials-12-00912-f005:**
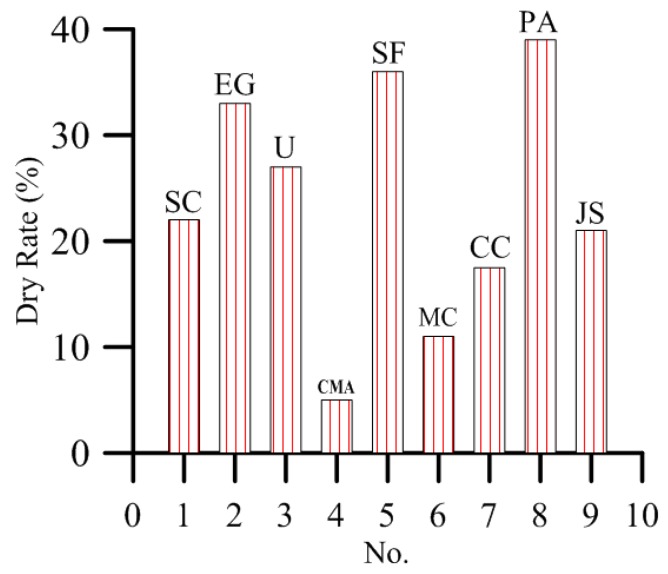
Dry rate of plants at seventh day under 3% concentration of de-icing salt.

**Figure 6 materials-12-00912-f006:**
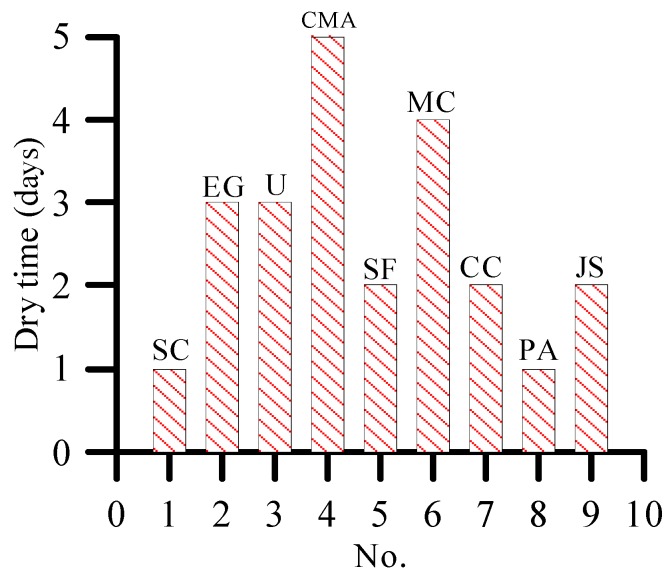
Plant tip yellowing days under 10% concentration of de-icing salt.

**Figure 7 materials-12-00912-f007:**
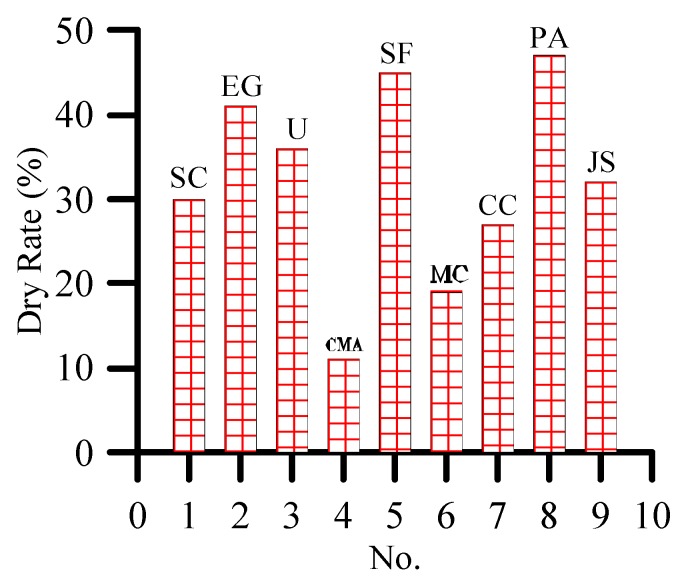
Dry rate of plants at seventh day under 10% concentration of de-icing salt.

**Figure 8 materials-12-00912-f008:**
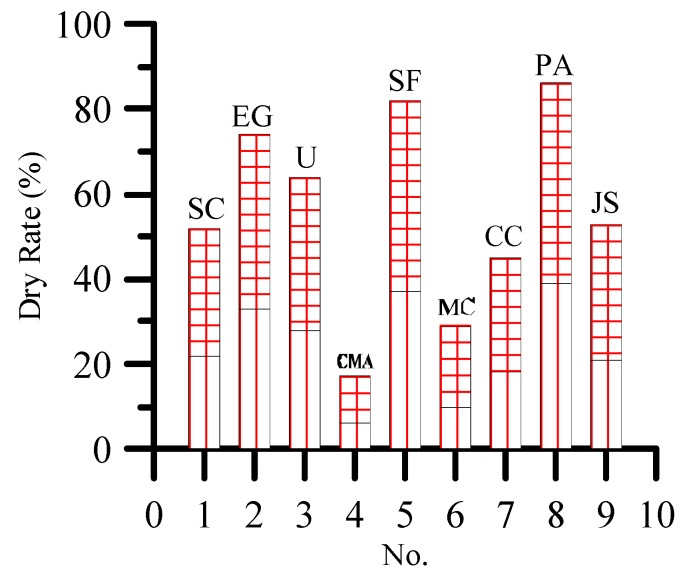
Dry rate of plants at seventh day under different concentrations of de-icing salt.

**Figure 9 materials-12-00912-f009:**
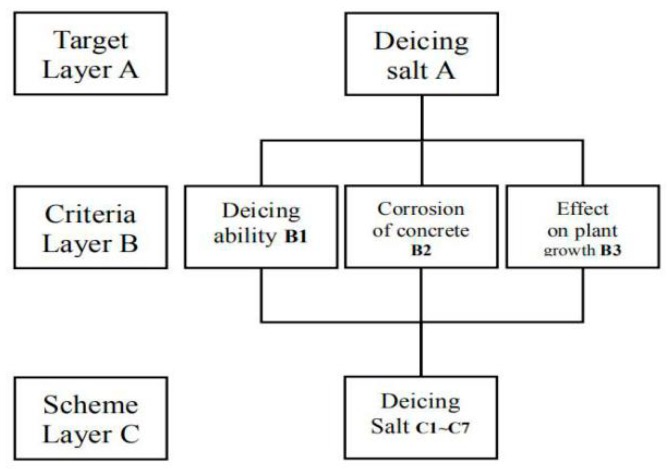
Hierarchical analysis structure of preferred choice of de-icing salt.

**Table 1 materials-12-00912-t001:** De-icing salt category.

No	1	2	3	4	5	6	7	8	9
Name	Sodium chloride (SC)	Ethylene glycol (EG)	Urea (U)	Calcium magnesium acetate (CMA)	Sodium formate (SF)	Magnesium chloride (MC)	Calcium chloride (CC)	Potassium acetate (PA)	Ji-suo II (JS)

**Table 2 materials-12-00912-t002:** De-icing ability test (18% concentration).

No	De-Icing Salt	De-Icing Accumulation/g					Baseline Percentage (30 min)/%
		30 min	60 min	90 min	120 min	150 min	
1	Sodium chloride	8.3	9.2	11.3	11.5	11.6	100.0
2	Ethylene glycol	8.7	10.3	12.0	12.2	12.2	104.8
3	Urea	7.6	8.8	9.6	9.7	9.9	91.6
4	Calcium magnesium acetate	7.9	9.0	9.8	10.4	10.5	95.2
5	Sodium formate	7.8	8.9	10.1	10.2	10.3	94.0
6	Magnesium chloride	8.0	9.3	10.0	10.4	10.6	96.4
7	Calcium chloride	8.5	9.7	10.1	10.3	10.4	102.4
8	Potassium acetate	5.8	6.6	7.5	7.7	6.8	69.9
9	Ji-suo II	9.7	11.1	10.9	7.8	−0.7	116.9

**Table 3 materials-12-00912-t003:** De-icing ability test (29% concentration).

No	De-Icing Salt	De-Icing Accumulation/g		
		30 min	60 min	90 min
1	Sodium chloride	15.75	18.71	17.13
2	Ethylene glycol	10.69	13.62	8.9
3	Urea	7.91	10.75	7.05
4	Calcium magnesium acetate	9.31	12.55	8.73
5	Sodium formate	—	—	—
6	Magnesium chloride	9.07	11.28	4.1
7	Calcium chloride	14.28	17.45	15.51
8	Ji-suo II	10.78	8.39	4.15

**Table 4 materials-12-00912-t004:** Peeling amount after 30 freeze–thaw cycles under different concentrations of de-icing salt.

No	De-Icing Salt	Concrete Peeling Amount (kg/m^2^)		
		3%	10%	20%
1	Sodium chloride	0.46	0.40	0.04
2	Ethylene glycol	0.52	0.44	0.12
3	Urea	0.57	0.42	0.00
4	Calcium magnesium acetate	0.28	0.19	0.00
5	Sodium formate	0.53	0.35	0.06
6	Magnesium chloride	0.14	0.20	0.00
7	Calcium chloride	0.41	0.23	0.10
8	Ji-suo II	0.49	0.38	0.05

**Table 5 materials-12-00912-t005:** Significance of scale.

Significance	Importance of a_i_ Relative to a_j_				
	The Same	Slightly Stronger	Stronger	Evidently Strong	Absolutely Strong
Scale a_ij_	1	3	5	7	9

**Table 6 materials-12-00912-t006:** Judgment Matrix A-B.

**(a)**				
**A**	**B1**	**B2**	**B3**	**W11**
B1	1	3	2	0.539
B2	1/3	1	1/2	0.164
B3	1/2	2	1	0.297
**(b)**				
**A**	**B1**	**B2**	**B3**	**W12**
B1	1	3	1	0.429
B2	1/3	1	1/3	0.142
B3	1	3	1	0.429

λ_max_ = 3.0055, CI = 0.003, RI = 0.58, CR = 0.006 < 0.10, Satisfactory consistency. λ_max_ = 2.9933, CI = −0.003, RI = 0.58, CR = −0.006 < 0.10, Satisfactory consistency.

**Table 7 materials-12-00912-t007:** Relative value calibration.

No	De-Icing Salt	B1/Relative Value Calibration	B2/Relative Value Calibration	B3/Relative Value Calibration
C1	Sodium chloride	100.0/6	100.0/3	100.0/5
C2	Ethylene glycol	104.8/9	113.3/2	150.0/1
C3	Urea	91.6/1	124.8/1	122.7/3
C4	Calcium magnesium acetate	95.2/3	60.8/7	22.7/9
C5	Sodium formate	94.0/2	116.1/1	163.6/1
C6	Magnesium chloride	96.4/4	31.5/9	50.0/8
C7	Calcium chloride	102.4/8	89.5/4	79.5/6

**Table 8 materials-12-00912-t008:** Judgment Matrix B1-C.

B1	C1	C2	C3	C4	C5	C6	C7	W2
C1	1	1/4	6	4	5	3	1/3	0.150
C2	4	1	9	7	8	6	2	0.381
C3	1/6	1/9	1	1/3	1/2	1/4	1/8	0.026
C4	1/4	1/7	3	1	2	1/2	1/6	0.054
C5	1/5	1/8	2	1/2	1	1/3	1/7	0.037
C6	1/3	1/6	4	2	3	1	1/5	0.079
C7	3	1/2	8	6	7	5	1	0.273

λ_max_ = 7.3089, CI = 0.051, RI =1.32, CR = 0.039 < 0.10, Satisfactory consistency.

**Table 9 materials-12-00912-t009:** Judgment Matrix B2-C.

B2	C1	C2	C3	C4	C5	C6	C7	W3
C1	1	2	3	1/5	3	1/7	1/2	0.076
C2	1/2	1	2	1/6	2	1/8	1/3	0.050
C3	1/3	1/2	1	1/7	1	1/9	1/4	0.032
C4	5	6	7	1	7	1/3	4	0.258
C5	1/3	1/2	1	1/7	1	1/9	1/4	0.032
C6	7	8	9	3	9	1	6	0.440
C7	2	3	4	1/4	4	1/6	1	0.111

λ_max_ = 7.2715, CI = 0.045, RI =1.32, CR = 0.034 < 0.10, Satisfactory consistency.

**Table 10 materials-12-00912-t010:** Judgment Matrix B3-C.

B3	C1	C2	C3	C4	C5	C6	C7	W4
C1	1	5	3	1/5	5	1/4	1/2	0.103
C2	1/5	1	1/3	1/9	1	1/8	1/6	0.027
C3	1/3	3	1	1/7	3	1/6	1/4	0.055
C4	5	9	7	1	9	2	4	0.376
C5	1/5	1	1/3	1/9	1	1/8	1/6	0.027
C6	4	8	6	1/2	8	1	3	0.268
C7	2	6	4	1/4	6	1/3	1	0.144

λ_max_ = 7.3276, CI =0.0546, RI = 1.32, CR = 0.041 < 0.10, Satisfactory consistency.

**Table 11 materials-12-00912-t011:** Average random consistency indicator RI value.

Matrix Order	1	2	3	4	5	6	7	8	9
RI	0	0	0.58	0.9	1.12	1.24	1.32	1.41	1.45

**Table 12 materials-12-00912-t012:** Hierarchical comprehensive sorting.

No.	B1(0.539/0.429)	B2(0.164/0.142)	B3(0.297/0.429)	W_total_1/W_total_2
C1	0.150	0.076	0.103	0.124/0.119
C2	0.381	0.050	0.027	0.222/0.182
C3	0.026	0.032	0.055	0.036/0.039
C4	0.054	0.258	0.376	0.183/0.221
C5	0.037	0.032	0.027	0.033/0.032
C6	0.079	0.440	0.268	0.194/0.211
C7	0.273	0.111	0.144	0.208/0.195

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
