# Peer review of "Evaluation and Selection of De-Icing Salt Based on Multi-Factor"

_materials, 2019, doi:10.3390/ma12060912_

Round 1
Reviewer 1 Report
All authors in this manuscript have made great efforts to describe the investigation of de-icing salts with 9 different types and grass plant and to evaluate the optimum de-icing salts with AHP process. Despite authors’ great efforts, there are a few specific issues the authors had better be addressed by making modifications to the manuscript or by clarifying in their response, after which I would consider this work suitable for publication and readers.
The introduction part should be added with recent papers and the purpose of this study should be mentioned clearly.
In page 3, grass species were used for the effect of de-icing slats on plant growth. Could you explain why the grass was selected for the effect of de-icing slats? Have you ever thought that actural street tree species were used to the experiment, instead of grass?
In page 8, AHP process has been usually applied to the decision making. For the readers, the detailed procedure and summary should be added in Figure 9 or additional figure.
Author Response
Sincerely thank the reviewer for your criticism and correction, thank you very much!
Point 1 | The introduction part should be added with recent papers and the purpose of this study should be mentioned clearly. |
Response 1:Thank you for your suggestion, I have updated it. | |
Point 2 | In page 3, grass species were used for the effect of de-icing slats on plant growth. Could you explain why the grass was selected for the effect of de-icing slats? Have you ever thought that actural street tree species were used to the experiment, instead of grass? |
Response 2:The selected grass species are mainly easy to operate, have certain representativeness, and have a short test period. I thought about using a normal tree species, but it took a lot of resources and a lot of time. | |
Point 3 | In page 8, AHP process has been usually applied to the decision making. For the readers, the detailed procedure and summary should be added in Figure 9 or additional figure. |
Response 3:Thank you for your suggestion, I have updated it. | |

Reviewer 2 Report
An interesting article. The conclusions from applying the decision making process sound interesting (less results as such, as the used variables are rather limited, but the procedure), however, in the current form too much focus is on presenting "old news". The article ends quite abrupt with a summary of the results and without stating its conclusions.
It is good, the authors try to explain exactly what they have done in their experiments. However, e.g. "dry rate" and "dry time" (figures 4-8) are not really explained but have to be guessed. Please explain the experiment better in the methods section.
Same is true for table 2, "de-icing accumulation" is insuficiently explained.
Line 103 "the same grass species" was used in the experiments. Please specifiy "grass" and/or explain that your choice of plants limits the transferability of your results (as different plants react differently to salt).
Line 165 calcium chloride magnesium instead of CMA
General comment on chapter 3 "Results": quite a lot of space is taken up by figures with relatively low information content. It would be better to summarize some of these and extend the explanation (results are difficult to understand, see above).
AHP: It is good that you explain your weighting, however, I could not follow it (maybe because I have no knowledge on AHP). If other reviewers had the same problem, maybe this section should be adapted.
Conclusions:
In its current state the "Conclusions" is only a summary of results. What are your main conclusions from your work?
The article deals with a topic where already a lot of research has been done. Among other, the effect on plant growth is only one aspect of environmental impacts of de-icers. So it should be explained why it was sufficient for the presented research question to limit the experiments to the chosen aspects.
The new insights achieved in this article should be presented more clearly as well as stating whether further research is needed (maybe by shifting its focus more to the decision making process?)
References
There is a good number of references included, however, why no review like e.g. https://link.springer.com/article/10.1007/s11270-011-1064-6
?
An interesting research but its presentation in the article should be improved.
Author Response
Sincerely thank the reviewer for your criticism and correction, thank you very much.
Point 1 | It is good, the authors try to explain exactly what they have done in their experiments. However, e.g. "dry rate" and "dry time" (figures 4-8) are not really explained but have to be guessed. Please explain the experiment better in the methods section. |
Response 1:Thank you for your suggestion, I have updated it. Dry day refers to the number of days when the tip of the plant begins to turn yellow; the dry rate refers to the proportion of plants that turn yellow. | |
Point 2 | Same is true for table 2, "de-icing accumulation" is insuficiently explained. |
Response2:Thank you for your suggestion, I have updated it. De-icing accumulation refers to the amount of ice that can be melted by deicing salt over a certain period of time. | |
Point 3 | Line 103 "the same grass species" was used in the experiments. Please specifiy "grass" and/or explain that your choice of plants limits the transferability of your results (as different plants react differently to salt). |
Response 3:Thank you for your suggestion, I have updated it. The selected grass species are mainly easy to operate, have certain representativeness, and have a short test period. | |
Point 4 | Line 165 calcium chloride magnesium instead of CMA |
Response 4:I have corrected it, thank you very much. | |
Point 5 | General comment on chapter 3 "Results": quite a lot of space is taken up by figures with relatively low information content. It would be better to summarize some of these and extend the explanation (results are difficult to understand, see above). |
Response 5:Thank you for your suggestion. The main focus of this article is the fourth chapter: the analytic hierarchy process is used to find a suitable snow melting agent. Therefore, the explanation in the chapter 3 is slightly less. | |
Point 6 | AHP: It is good that you explain your weighting, however, I could not follow it (maybe because I have no knowledge on AHP). If other reviewers had the same problem, maybe this section should be adapted. |
Response 6:Thank you for your suggestion. The weight value is strictly derived and calculated according to the AHP method, which is similar to the actual user response, indicating that the method can be used well. | |
Point 7 | Conclusions:In its current state the "Conclusions" is only a summary of results. What are your main conclusions from your work? |
Response 7:Thanks for your suggestion. The main goal of this paper is to find an optimal deicing salt meets the requirements through analytic hierarchy process, and provide a method. | |
Point 8 | The article deals with a topic where already a lot of research has been done. Among other, the effect on plant growth is only one aspect of environmental impacts of de-icers. So it should be explained why it was sufficient for the presented research question to limit the experiments to the chosen aspects. |
Response 8:Thanks for your suggestion. The most important thing in this article is to provide a method. Of course, the choice of deicing salt should take into account other factors, which is subjective. I think the impact on plants and concrete is the most important and can be used as an object to study this method. | |
Point 9 | The new insights achieved in this article should be presented more clearly as well as stating whether further research is needed (maybe by shifting its focus more to the decision making process?) |
Response 9:Thanks for your suggestion. The new insight of this paper is mainly to solve the problem of deicing salt selection by using analytic hierarchy process. | |
Point 10 | References There is a good number of references included, however, why no review like e.g. https://link.springer.com/article/10.1007/s11270-011-1064-6 |
Response 10:I have updated it, thank you very much. | |
